# Changes in Health Care Access during the COVID-19 Pandemic: Estimates of National Japanese Data, June 2020–October 2021

**DOI:** 10.3390/ijerph19148810

**Published:** 2022-07-20

**Authors:** Yuta Tanoue, Cyrus Ghaznavi, Takayuki Kawashima, Akifumi Eguchi, Daisuke Yoneoka, Shuhei Nomura

**Affiliations:** 1Department of Health Policy and Management, School of Medicine, Keio University, 35 Shinanomachi, Shinjyuku-ku, Tokyo 160-8582, Japan; tanoue.yuta@aoni.waseda.jp (Y.T.); cghaznavi@keio.jp (C.G.); kawashima@c.titech.ac.jp (T.K.); siero5335@gmail.com (A.E.); blue.sky.sea.dy@gmail.com (D.Y.); 2Institute for Business and Finance, Waseda University, 1-6-1 Nishi-Waseda, Shinjuku-ku, Tokyo 169-8050, Japan; 3Medical Education Program, Washington University School of Medicine, 660 S Euclid Ave, Saint Louis, MO 63110, USA; 4Department of Mathematical and Computing Science, Tokyo Institute of Technology, 2-12-1 Ookayama, Meguro-ku, Tokyo 152-8550, Japan; 5Center for Preventive Medical Sciences, Department of Sustainable Health Science, Chiba University, 1-33 Yayoi-cho, Inage-ku, Chiba 263-8522, Japan; 6The Tokyo Foundation for Policy Research, 3-2-1 Roppongi, Tokyo 106-6234, Japan; 7Infectious Disease Surveillance Center, National Institute of Infectious Diseases, 1-23-1 Toyama, Tokyo 162-8640, Japan; 8Department of Global Health Policy, Graduate School of Medicine, The University of Tokyo, 7-3-1 Hongo, Bunkyo-ku, Tokyo 113-8654, Japan

**Keywords:** COVID-19, health care access, healthcare capacity, hospital beds, hospital stay, outpatient visit, Japan

## Abstract

The COVID-19 pandemic has disrupted health care access around the world, both for inpatients and outpatients. We applied a quasi-Poisson regression to national, monthly data on the number of outpatients, number of inpatients, length of average hospital stay, and the number of new hospitalizations from March 2015 to October 2021 to assess how these outcomes changed between June 2020 to October 2021. The number of outpatient visits were lower-than-predicted during the early phases of the pandemic but normalized by the fall of 2021. The number of inpatients and new hospitalizations were lower-than-predicted throughout the pandemic, and deficits in reporting continued to be observed in late 2021. The length of hospital stays was within the predicted range for all beds, but when stratified by bed type, was higher than predicted for psychiatric beds, lower-than-predicted for tuberculosis beds, and showed variable changes in long-term care insurance beds. Health care access in Japan was impacted by the COVID-19 pandemic.

## 1. Introduction

### 1.1. Background

The COVID-19 pandemic has placed undue strain on health care systems around the world [1], and hospitals have been forced to make difficult decisions regarding the number of patients that can be admitted, the length of patient stays, and the preparation of COVID-19-dedicated beds. During the highest intensity periods of COVID-19 transmission, elective surgeries were routinely cancelled and waiting times were increased in order to conserve beds [2,3,4], and routine clinic visits such as those for vaccination [5], cancer screenings [6,7,8], and sexually transmitted disease testing [9] were often foregone [10]. In fact, the fear of contracting COVID-19 dissuaded many from even presenting to health care establishments [11,12], leading to breaks in the continuity of patient care, decreased preventive medicine, and delayed diagnosis and management of otherwise treatable medical conditions [13]. These collateral effects of the pandemic are often measured using excess deaths, which have been higher-than-expected in multi-country analyses attempting to quantify the unseen impacts of COVID-19 [14].

Unlike many Western nations facing overwhelming COVID-19 notifications and inflated excess deaths during the early phases of the pandemic in 2020, Japan saw only low-levels of transmission and paradoxically observed lower-than-expected (i.e., exiguous) deaths [15]. The lessons behind this success are drawing worldwide attention [16], and it was probably due in part to a national state of emergency declaration (SoE) that was implemented before uncontrolled transmission in April–May 2020. During the SoEs, while protective measures such as mask wearing were boosted, businesses (e.g., restaurants and bars) were encouraged to abide by curfews to limit social gatherings in the evening, and individuals were encouraged to work from home and minimize outings, except for those that were truly necessary [17]. Unlike in many other countries, the SoE provisions in Japan were voluntary (as Japan’s constitution forbids enforcement) and the government struggled with messaging [16]; though the mobility data suggest that many followed these precautions, compliance decreased with each subsequent declaration [18].

A previous analysis of health care access in Japan, using data available until June 2020, found that there was a decreased number of outpatients and hospitalized patients as well as a slight drop in the length of the average hospital stay that largely coincided with the first SoE [19]. Notably in 2020, not all health care institutions accepted COVID-19 patients, and many institutions maintained pre-pandemic functioning by cordoning COVID-19 patients to a certain subset of hospitals. Medical institutions in Japan are not legally obliged to accept COVID-19 patients, and national and local governments cannot force them to do so. Some hypothesize that this partitioning of COVID-19 vs. non-COVID-19 hospitals may have contributed somewhat to minimizing the impact of the COVID-19 pandemic on the Japanese health care system for other diseases. If true, while there is no perfect solution that can be universally applied in all countries as part of the COVID-19 pandemic response, the state of the Japanese health care system during the pandemic may provide lessons that benefit other countries. It is therefore important to understand how health care access changed over time in Japan, especially during the SoEs and after their liftings. Further data are needed to assess whether certain types of care (e.g., outpatient vs. inpatient) were differentially affected by the pandemic, and whether they have recovered in tandem or asynchronously.

COVID-19 transmission increased substantially the following year: compared to 234,109 new cases in 2020, 1,492,904 new cases were reported in 2021 [20]. Large waves of infection were noted in January, May, and August 2021, which prompted up to three more SoEs in Japan’s varying 47 prefectures. Consequently, Japan also experienced excess—as opposed to exiguous—deaths in 2021 [21,22]. Even after excluding deaths due to COVID-19, respiratory system-related illnesses, and circulatory system-related illnesses, deaths were still found to be in slight excess in some prefectures, suggesting that previous levels of health care access may have deteriorated in the face of overwhelming COVID-19 case numbers [22].

Among the OECD nations, Japan has some of the highest numbers of hospital beds per population and length of hospital stay [23,24]; in combination with the far-reaching effects of universal health insurance on health care utilization, Japanese residents benefit from some of the best health care access in the world [25]. In this sense, Japan is a valuable case study of health care access during the pandemic era and can serve as a useful model for other nations. Assessments of when (or if) Japanese health care access returns to pre-COVID-19 baselines may represent a theoretical “lower bound” for other countries where health care access was traditionally more restricted in pre-pandemic times.

### 1.2. Objectives

We hypothesize that Japan’s health care utilization never fully recovered from the initial insult of the pandemic on health care access. The sudden reversal of exiguous to excess deaths in Japan comprises a unique opportunity to evaluate which factors underlie excess mortality during the pandemic era. In this study, we used national, monthly data on the number of outpatients, number of inpatients, length of average hospital stays, and the number of new hospitalizations through October 2021 to assess how health care utilization changed in Japan during the early and late phases of the pandemic. Specifically, we evaluate whether health care access, stratified by the type of utilization, was in excess or deficit for any given month. The findings of our study will serve as a sentinel metric of the state of health care access in Japan during the late phases of the COVID-19 pandemic and will aid policymakers in designing measures to encourage appropriate health care access during the pandemic era.

## 2. Materials and Methods

### 2.1. Data

As in previous studies [19], this analysis was conducted using hospital report data from the Ministry of Health, Labor, and Welfare, which has been published since 1949. The openly-available dataset includes monthly data on the number of in/outpatients, length of hospital stays, and the number of hospitalizations and discharges, stratified by prefecture. The purpose of the hospital report is to obtain basic data for the formulation of health care policy by understanding the current status of patient utilization and health care professionals at hospitals and clinics with medical care beds throughout Japan [26]. Outpatient visit counts were available for general hospitals, psychiatric hospitals, and in total. Inpatient statistics were available stratified by bed type: psychiatric, tuberculosis, long-term care, general, long-term care covered by long-term care insurance (LTCI: included in long-term care), and in total. The present study utilizes nationally aggregated, monthly data from March 2015 to October 2021 for the number of outpatients, number of inpatients, length of average hospital stays, and the number of new hospitalizations; predictions were estimated for the period between June 2020 and October 2021.

### 2.2. Statistical Approach

The analysis in this study was conducted using the Farrington algorithm, a method commonly used to assess excess deaths during the COVID-19 pandemic or to flag outbreaks by the Centers for Disease Control and Prevention in the U.S. [27,28].

The Farrington algorithm is a quasi-Poisson regression model that uses a moving window of historical data, in addition to considerations for seasonality, to construct a baseline for future predictions [29,30]. We estimated the expected value of all outcome variables for any given month as well as the two-sided 95% prediction interval for each value. The quasi-Poisson regression used in the Farrington algorithm can be expressed as:(1)$$log\left(E\left(Y_{t}\right)\right)=\alpha +\beta t\pm f^{T}\left(t\right)\gamma_{f}\;var\left(Y_{t}\right)=\Phi E\left(Y_{t}\right)$$

For the purposes of estimating parameters, we considered time-point $t=\left(a:A\right)$, where *a* and *A* represent a given week in a year and a given year, respectively. For $t=\left(a:A\right)$, we call the set of time-points, $\left\{a-w,\;a-w+1,\;\ldots ,\;a+w-1,\;a+w\right\}$, in the years $\left\{A-B,\;A-B+1,\;\ldots ,\;A-1\right\}$ and {a−w, …, a−1} in year $\left\{A\right\}$ as the reference periods. Furthermore, we divided each time period between each reference period into n equally sized periods. The Farrington algorithm only uses data recorded from time-point $\left(a-w:A-B\right)$ to time-point $\left(a-p:\;A\right)$ to estimate the regression parameters for time-point $t=\left(a:A\right)$. In this study, we set w=3, B=5, and n=3. Because the first case of COVID-19 was identified in Japan during January 2020, we limited the use of historic data until December 2019 when estimating the expected values and prediction intervals. Thus, we used *p* = 26 until June 2020, after which we adaptively changed the value of *p* so that data from January 2020 and beyond were not used for the estimation.

In Equation (1), $Y_{t}$ is the outcome value at time  t; α, β, and $\gamma_{f}$ are regression parameters; and $f^{T}$ is a vector of dummies that indicates which period (reference period or equally-sized period outside of the reference period) to which time-point *t* belongs. The terms α and β capture long-term trends. The term $f^{T}\left(t\right)\gamma_{f}$ captures seasonality. Parameters including the overdispersion parameter, Φ, are estimated by the quasi-likelihood approach. The average length of hospital stay outcome variable was rounded to the nearest integer so that the use of a quasi-Poisson-based regression procedure was mathematically possible.

Once the regression parameters for each time-point *t* were estimated, the expected value was predicted for that time-point. The two-sided 95% prediction intervals were then estimated by assuming that the data followed the negative binomial distribution as $Y_{t}∼NB\left(\hat{Y_{t}},\hat{v_{0}}\right)$, where $\hat{Y_{0}}$ is the mean of the distribution and $\hat{v_{0}}=\frac{\hat{Y_{t}}}{\Phi -1}$ is the dispersion parameter. Any observed value above the upper bound or below the lower bound of the 95% prediction interval was referred to as a statistically significant excess or deficit, respectively. All calculations were performed in *R* version 4.1.1 using the *surveillance* package.

## 3. Results

### 3.1. Number of Oupatients

The monthly average number of outpatients per day at general hospitals, psychiatric hospitals, and in total are shown in Figure 1 and Appendix A. Statistically significant deficits in the number of outpatients at general hospitals were detected in June to September 2020, November 2020, January to February 2021, May 2021, and July 2021. Statistically significant deficits in the number of outpatients at psychiatric hospitals were detected in August 2020, November 2020, January 2021, and May 2021. Statistically significant deficits in the number of total outpatients were detected in June to September 2020, November 2020, January to February 2021, May 2021, and July 2021. No statistically significant excesses were detected during the study period.

### 3.2. Number of Inpatients

The monthly average number of inpatients per day in psychiatric beds, tuberculosis beds, long-term care beds, general beds, LTCI beds, and in total are shown in Figure 2 and Appendix A. Statistically significant deficits in the number of inpatients in psychiatric beds, long-term care beds, general beds, LTCI beds, and total beds were detected over the entire study period (June 2020 to October 2021). Statistically significant deficits in the number of inpatients in tuberculosis beds were detected in June 2020 and January to October 2021.

### 3.3. Hospital Stay

The monthly average hospital stay (in days) for psychiatric beds, tuberculosis beds, long-term care beds, general beds, LTCI beds, and in total are shown in Figure 3 and Appendix A. Statistically significant excesses in the hospital stay for psychiatric beds were detected in January 2021 and May 2021. Statistically significant deficits in the hospital stay for tuberculosis beds were detected in July to August 2020, December 2020, May 2021, and July to September 2021. No statistically significant deficits or excesses in the hospital stay for long-term care beds or general beds were detected. Statistically significant excesses in the hospital stay for LTCI beds were detected in October 2020 to March 2021 and May 2021; a significant deficit was detected in September 2021. No statistically significant deficits or excesses in the hospital stay for all beds were detected.

### 3.4. New Hospitalizations

The monthly number of new hospitalizations for psychiatric beds, tuberculosis beds, long-term care beds, general beds, LTCI beds, and in total are shown in Figure 4 and Appendix A. Statistically significant deficits in new hospitalizations for psychiatric beds were detected in November 2020, January to February 2021, May 2021, and July to September 2021. Statistically significant excesses in new hospitalizations for tuberculosis beds were detected in June to August 2020, December 2020 to January 2021, March to May 2021, and July to September 2021; a significant deficit was detected in February 2021. Statistically significant deficits in new hospitalizations for long-term care beds were detected in June to September 2020, November 2020, January to February 2021, and April to October 2021. Statistically significant deficits in new hospitalizations for general beds were detected in June to September 2020, November 2020 to February 2021, and April to October 2021. Statistically significant deficits in new hospitalizations for LTCI beds were detected over the entire study period (June 2020 to October 2021). Statistically significant deficits in new hospitalizations for all beds were detected in June to September 2020, November 2020 to February 2021, April to July 2021, and September to October 2021.

## 4. Discussion

Using the Japanese national hospital report data, we assessed the changes in the number of outpatients, inpatients, days of average hospital stay, and new hospitalizations during the COVID-19 pandemic. We found that the outpatient visits decreased broadly during the early phases of the pandemic but largely returned to the within-predicted-range by August to October 2021. The number of inpatients and new hospitalizations suffered from significant deficits both early and late in the pandemic. Finally, the length of hospital stays was unaffected for all beds, but when stratified by bed type, it showed some increases among the psychiatric and LTCI beds and decreases among the tuberculosis beds.

Our findings are consistent with an initial decrease in outpatient visits to general and psychiatric hospitals as well as in total, during the early stages of the pandemic, followed by alternating cycles of brief returns to normality and deficits aligning with the new SoEs. Ultimately, outpatient visits rebounded to within-predicted range during the last three months of the study period (August to October 2021). The reasons for the decrease in visits are two-fold. First, many individuals may have avoided visiting clinical spaces out of concern that they may be exposed to SARS-CoV-2 [12]. In fact, a survey of 30,053 Japanese individuals in February 2021 found that up to 35.9% of respondents avoided visiting a medical institution because they were anxious about contracting COVID-19 [11]. Alternatively, the consistent decreases in outpatient visits during the SoEs may reflect not only clinic avoidance out of fear of contracting COVID-19, but also a general reluctance to leave one’s home in accordance with local government requests to minimize outings during periods of high transmission. Second, many (but not all) hospitals imposed limits on the number of outpatient appointments (or closed outpatient departments altogether) due to COVID-19-related staff shortages, and in an attempt to minimize viral exposure among outpatients and health care workers [31]. These findings are consistent with widespread reports of decreased routine clinic visits including those for routine vaccinations, cancer screenings, and STI testing [5,6,7,8,9].

The number of inpatients and new hospitalizations showed similar trends, largely consisting of significant deficits since the start of the pandemic. With the exception of tuberculosis, the decreased number of inpatients and new hospitalizations is likely attributable to three distinct phenomena. First, much like with outpatients, inpatient admission restrictions were implemented [31]. Even when hospitals were willing to accept patients, hospital infection prevention programs limited the number of those who could be accepted in order to suppress the risk of nosocomial COVID-19 transmission. Second, the prefectural governments called on medical institutions to establish inpatient treatment systems for COVID-19 patients [32], but it was left to the discretion of each medical institution as to whether to actually accept them, resulting in high concentrations of COVID-19 patients at limited institutions while there was relatively normal functioning at the remaining clinical spaces [33]. Finally, compared to many other OECD nations, Japan has many more beds per population (12.8 [Japan] vs. 4.4 [OECD average] per 1000 people in 2019 including all types of beds) [23], though the number of doctors/nurses per population is on par with or lower than in European nations. In contrast, the average length of hospital stay for acute care in Japan is much longer than that of the OECD average (16.0 days vs. 6.6 days in 2019) [24], which has been touted as evidence of possible inefficiencies in the Japanese health care system. Thus, there is a belief that the threshold for admitting patients in Japan is relatively low because there is an abundance of beds, even accounting for the fact that the Japanese population is aging more rapidly than in other countries. During the COVID-19 pandemic, we believe that many institutions raised the threshold for patient admission in an abundance of caution to maintain hospital capacity. Thus, many patients that would have been admitted pre-COVID-19 were likely to be sent home with supervision and hospital return instructions, resulting in lower inpatient numbers. These findings are consistent with reports suggesting that the number of surgeries in Japan decreased during the pandemic [34].

Tuberculosis beds were a notable exception, however, with decreased numbers of inpatients but increased numbers of new hospitalizations. The decreased number of inpatients in likely attributable to the fact that rates of tuberculosis decreased in Japan during the pandemic [35]. Increased social distancing and masking decreased the aerosol transmission of tuberculosis, and decreased travel because of tight border restrictions essentially eliminated imported cases of tuberculosis. Furthermore, tuberculosis patients being cared for in the hospital may have been sent home in an attempt to minimize exposure to COVID-19 patients being held in the same infectious disease unit, since coinfection with COVID-19 would result in significant morbidity and mortality [36]. Notably, the increase in the number of admissions to tuberculosis beds observed throughout the pandemic was likely the result of tuberculosis beds being siphoned off for use by COVID-19 patients [37]. Though these beds were allocated to non-tuberculosis patients, their use was still counted as tuberculosis beds in official hospital statistics. Both these reasons may also explain why the average hospital stay for tuberculosis patients was found to be significantly lower than expected during the pandemic. In addition, the fact that COVID-19 patients were being treated in tuberculosis beds also likely brought down the average hospital stay, as tuberculosis treatment may last longer on average than non-ICU COVID-19 treatment.

The number of inpatients and new admissions to LTCI beds showed significant deficits long before the pandemic began, though these deficits were exacerbated in May 2020, during the time of the first SoE. The use of LTCI beds has tapered off as part of a national policy to abolish LTCI beds in hospitals and instead shift patients to nursing homes by the end of March 2024 [38]. Thus, the trends seen in LTCI beds are not only attributable to the pandemic, though COVID-19 seems to have accelerated the rate at which the use of these beds has decreased. It is likely that many of the elderly patients who were using these beds were encouraged to go home to decrease their risk of in-hospital exposure to COVID-19 [19]. Of note, LTCI beds were included in the number of long-term care beds, which is why the trends for these two categories resemble one another.

Our analysis of the average length of hospital stays per month found that though in total there were no changes during the pandemic, hospital stays increased for psychiatric beds, and then decreased and then increased for the LTCI beds. In the case of psychiatric beds, the significant increase in hospital stays aligned with the SoE measures, which suggests that there may have been difficulties with disposition planning and patient discharges [39]. As many psychiatric patients may hail from vulnerable social situations (e.g., poor housing, complicated family dynamics, etc.), they would have been particularly susceptible to poor discharge circumstances during the worst periods of the pandemic in Japan. Notably, the average hospital stay increased by about one month for psychiatric beds in May 2020, suggesting that discharges may have been pending until the end of the SoE. In contrast, hospital stays for LTCI beds decreased from April to May 2020, possibly because patients were being rushed out of the hospital in an attempt to minimize their exposure to COVID-19 patients during the first surge. However, this sudden increase in the number of LTCI patients being sent to nursing homes likely led to the decreased capacity to discharge patients later in the pandemic, resulting in the increased hospital stays observed in late 2020 and early 2021.

Currently, the Omicron BA.4 and BA.5 variants are gaining momentum in Japan, and are even more transmissible than the previous BA.1 and BA.2 variants. Although vaccination coverage is high in Japan, breakthrough infections and re-infections have been reported, as has already been observed in other countries [40]. In fact, the number of COVID-19 infections in Japan has increased dramatically since the beginning of July 2022, but there is currently no solid evidence that the BA.4 and BA.5 variants cause more severe illness than the previous variants. Future trends in the number of infections and serious cases and their impact on the health care system are difficult to predict. However, at the same time, as in other countries, Japan is now lifting various behavioral restrictions, and as the reality of COVID-19’s conversion from a pandemic to an endemic settles, we must formulate health policies that allow us to adapt to the new normal. When considering appropriate health system functioning in the COVID-19 era, we must first delineate how the pandemic has changed the typical patterns of health care access and whether those changes are sustainable. We encourage policymakers to consider our findings when crafting new policies aimed at encouraging appropriate health care access, for example, prefecture-level decisions to expand hospital bed capacity during COVID-19 peaks and to request medical institutions accept COVID-19 patients. Furthermore, Japan serves as a valuable case study for other nations that may have struggled to maintain hospital access during the pandemic. Considering Japan’s relatively low COVID-19 infection rate, the trends and patterns seen in Japanese health care access, which are bolstered by the universal health insurance system, may serve as an optimistic prediction for other countries where health care access has traditionally been more restricted.

This study has some limitations. First, the Farrington algorithm uses historical data to construct a model that predicts future trends, which are then compared to observed data. In so doing, the Farrington algorithm accounts for gradual changes over time, but cannot absorb sudden changes such as those due to policy. Thus, the interpretation of our findings is limited by potential confounders that may have been at play during the study period (e.g., the reduced number of LTCI beds due to national policy changes or other efforts that began prior to the pandemic to change the distribution of hospital beds or length of hospital stays). Second, the outpatient data used in this study were limited to visits to in-hospital clinics. However, unlike many other high-income nations where outpatient visits primarily occur in non-hospital clinics, the use of in-hospital clinics is widespread in Japan and thus merits attention. Third, as these data were based on the clinic and hospital reporting of visit tabulations and bed usage among all Japanese residents, our findings are subject to reporting bias. Finally, our findings allow us to make conjectures as to why the observed values may differ from the predicted values, but we cannot make definitive assessments of causality using this approach.

## 5. Conclusions

The COVID-19 pandemic has changed the way people access health care around the world including in Japan. The use of outpatient services was affected by the SoEs during the early- and mid-pandemic periods, but ultimately rebounded to normal in late 2021. In contrast, the use of inpatient services has decreased since the first SoE was declared in April 2020, which continued at least through to October 2021, the end of the study period. As far as 2020 is concerned, when no excess deaths were observed, the fact that the acceptance of COVID-19 patients was limited to a certain subset of hospitals may have resulted in minimizing the impact of the COVID-19 pandemic on the Japanese health care system for other diseases. However, the finding of excess deaths during the 2021 COVID-19 waves suggests that the hospitals have not yet fully adjusted to the pandemic situation. It remains unclear whether returning to pre-COVID-19 levels of hospital functioning in Japan would be beneficial, as the Japanese health care system has been previously criticized as being less efficient than that of many other high-income nations. The changing use of health care in Japan is also bound to have long-term health effects for patients with respect to preventive medicine, continuity of care, and management of severe illness. Research is needed to elucidate the mechanisms linking these changes in the use of health care and the already observed excess deaths, and to evaluate their association with future long-term health outcomes.

## Figures and Tables

**Figure 1 ijerph-19-08810-f001:**
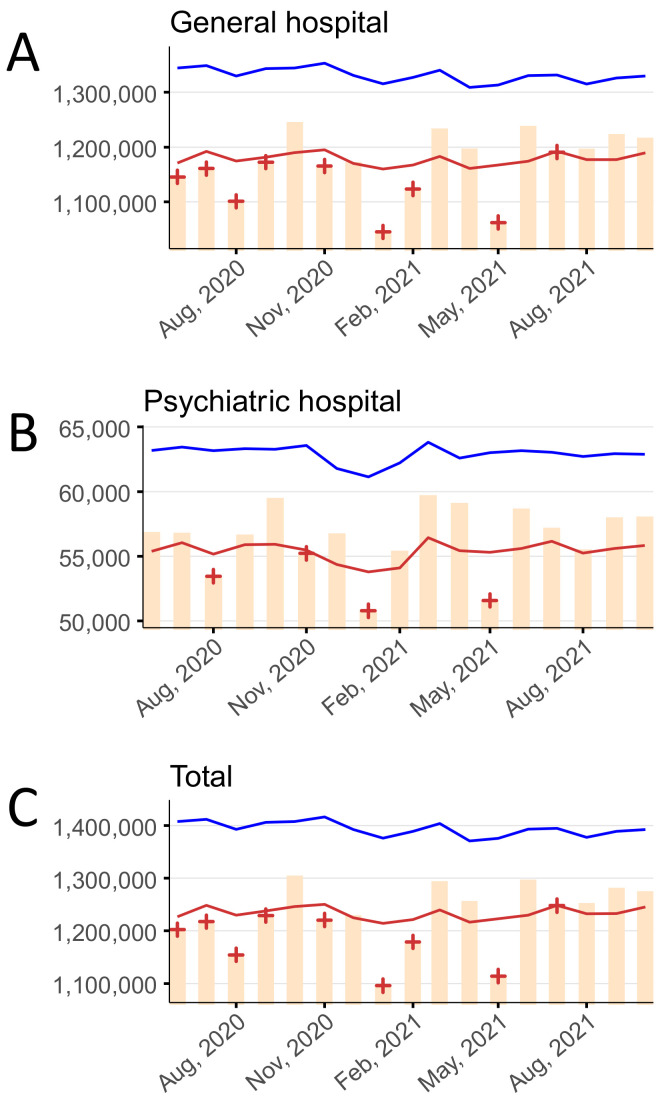
The monthly average number of outpatients per day stratified by hospital type, June 2020–October 2021 in (**A**) general hospitals, (**B**) psychiatric hospitals, and (**C**) in total. Yellow bars indicate the observed number of outpatients per day. Blue and red lines show the upper and lower limits, respectively, of the 95% prediction interval. Red + symbols indicate statistically significant monthly deficits. Gray shading corresponds to the state of emergency declarations in Tokyo.

**Figure 2 ijerph-19-08810-f002:**
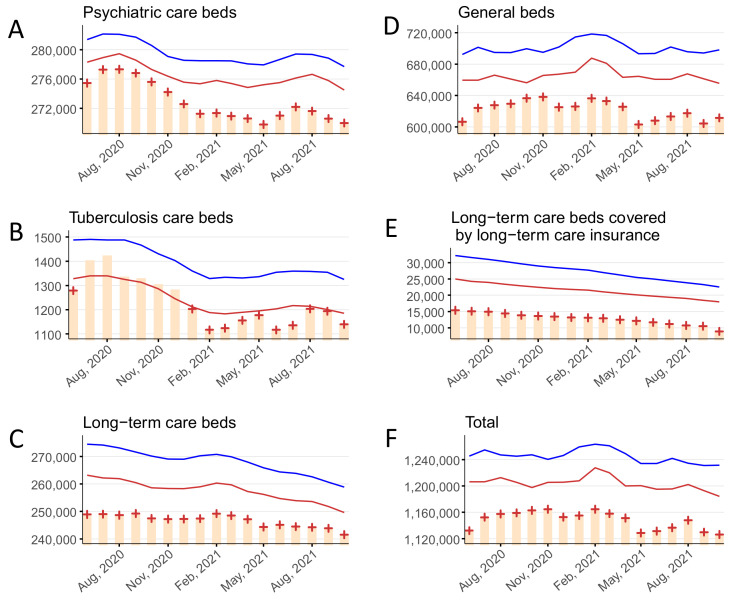
The monthly average number of inpatients per day stratified by bed type, June 2020–October 2021, in (**A**) psychiatric beds, (**B**) tuberculosis beds, (**C**) long-term care beds, (**D**) general beds, (**E**) long-term care beds covered by long-term care insurance, and (**F**) in total. Yellow bars indicate the observed number of inpatients per day. Blue and red lines show the upper and lower limits, respectively, of the 95% prediction interval. Red + symbols indicate statistically significant monthly deficits. Gray shading corresponds to the state of emergency declarations in Tokyo.

**Figure 3 ijerph-19-08810-f003:**
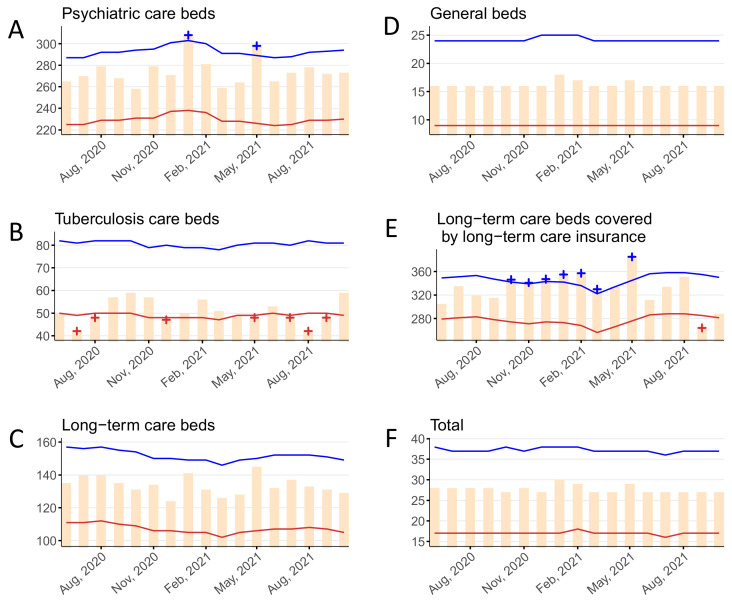
The monthly average length of hospital stay per patient stratified by bed type, June 2020–October 2021 in (**A**) psychiatric beds, (**B**) tuberculosis beds, (**C**) long-term care beds, (**D**) general beds, (**E**) long-term care beds covered by long-term care insurance, and (**F**) in total. Yellow bars indicate the observed number of days. Blue and red lines show the upper and lower limits, respectively, of the 95% prediction interval. Blue and red + symbols indicate statistically significant monthly excesses and deficits, respectively. Gray shading corresponds to the state of emergency declarations in Tokyo.

**Figure 4 ijerph-19-08810-f004:**
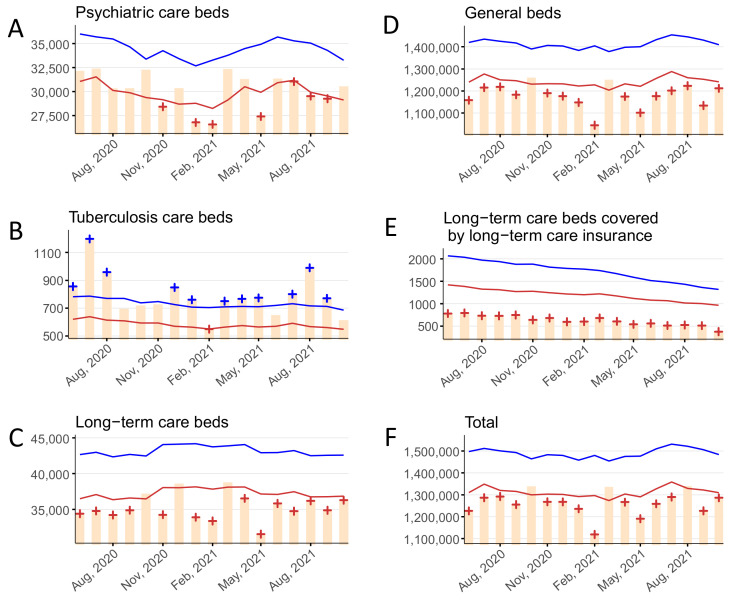
The monthly number of new hospitalizations stratified by bed type, June 2020–October 2021 in (**A**) psychiatric beds, (**B**) tuberculosis beds, (**C**) long-term care beds, (**D**) general beds, (**E**) long-term care beds covered by long-term care insurance, and (**F**) in total. Yellow bars indicate the observed number of new hospitalizations per month. Blue and red lines show the upper and lower limits, respectively, of the 95% prediction interval. Blue and red + symbols indicate statistically significant monthly excesses and deficits, respectively. Gray shading corresponds to the state of emergency declarations in Tokyo.

## Data Availability

The data are publicly available at https://www.mhlw.go.jp/toukei/list/79-1a.html accessed on 17 July 2022.

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
