# Peer review of "Changes in Health Care Access during the COVID-19 Pandemic: Estimates of National Japanese Data, June 2020–October 2021"

_ijerph, 2022, doi:10.3390/ijerph19148810_

Round 1
Reviewer 1 Report
Tanoue Y et al. analyzed trends in healthcare access in Japan during the COVID-19 pandemic using the openly-available hospital report data in Japan. The present reviewer thinks this manuscript is well written, and the analyses and topics of the present study are interesting. In addition, the limitations of the present study are appropriately described in the Discussion section. The present reviewer thinks this manuscript is worth publishing in IJERPH. I have comments as follows.
Line 322: The authors described that the interpretation of the Farrington algorithm is limited by confounders that may have been at play during the study period. It would be better if authors describe specific and possible confounders that may have affected the interpretation of their study results.
Reviewer 2 Report
Attached

Reviewer 3 Report
The material is interesting and the topic is relevant. The method seems to have been followed faithfully and the authors were well-positioned to conduct the analysis. The paper is well-written and the analysis is straight-forward and robust. Despite these positives, in my view, the paper needs more work before it could be published and I have made some specific suggestions below.
- The literature addressed is not described accurately so far as I can see. Relevant literature should be presented more deeply in order to support the research problem. Further, there is no clear distinction between manuscript sections in terms of the content they report. First, I suggest dividing the section "Introduction" into three components, respectively introduction (explain the general argument of the paper, without going into specific details) background (situate the study concepts within the context of extant knowledge, discuss the international relevance of the concepts) and purpose, creating greater clarity in the analysis of the reader. What is the study's biggest contribution? The contribution should be clearly stated in the introduction.
- Description of the social restrictions - lockdowns - in Japan and how these correlate to hospital activity.
Data Collection
- It is important to clarify more deeply the purpose of the original dataset (Japanese national hospital report data) because this can influence many factors such as the targeted population, the sample selected and the general context of the study.
- The ethical aspects of collecting data are not specifically clarified. If all the data protection provisions on pseudo-anonymization of all personal data are fulfilled and no link to primary data is intended.
- Why does the study terminate in Oct 2021? It would be very useful to see what happened to hospital activity over subsequent months - did activity stay depressed in the end of 2021 or return to normal?
Results
- It would be important to disaggregate data into elective / planned versus emergency hospital admissions - there could be different patterns.
- There are a large number of charts. It is very difficult to interpret data from some charts due to their small size and overlapping lines. Please provide all underlying data in tabulated form in an appendix so that readers can easily access this.
- The anonymised nature of the database means that the hospitals’ location was not disclosed. Therefore, because the infection status of each hospital and their surrounding region are unknown, it is possible that some of the hospitals limited services for non-COVID-19 patients so that they could treat COVID-19 patients or because of nosocomial COVID-19 infections." It's not clear why aggregated multi-centre data requires anonymisation of hospitals and therefore loss of fidelity over local COVID rates - it should be explained.
Discussion
- The recommendations should have been approached in greater depth. As the pandemic continues to unfold, the authors are encouraged to strengthen the discussion of implications of the study results for policy and practice.
CHECKLIST FOR STYLE
Organization and style: The manuscript is clearly written and will serve a broad audience of students, researchers, and practitioners. References should be prepared according to the ACS style guide.
Reviewer 4 Report
The subject is interesting considering COVID19 data. But
the theoretical analysis is very poor without any significant information about the theory, modeling and comparions.
The authors have given a lot of effort to analyse their data. Inside their paper are given some statistical notes without any statistical analysis or explanations (for example lines 161-162 Statistically significant excesses in the number of inpatients. From where this result is given based any statistical methodology?
In general the analysis of the authors is not explained properly without statistical evidence. Same point have been recognised in different lines.
If the authors want to publish their work, they must analyse their methodology with more information and use statistical methodology to justify their results
Round 2
Reviewer 2 Report
I think the manuscript is ok now
Reviewer 3 Report
I believe that the review carried out has greatly improved the quality of the study. Also, I do think that the author(s) addresses the broad questions, appropriately which were asked. Congratulations!
Reviewer 4 Report
Accept